# Increased Methylation of Brain-Derived Neurotrophic Factor (*BDNF)* Is Related to Emotionally Unstable Personality Disorder and Severity of Suicide Attempt in Women

**DOI:** 10.3390/cells12030350

**Published:** 2023-01-17

**Authors:** Esmail Jamshidi, Adrian E. Desai Boström, Alexander Wilczek, Åsa Nilsonne, Marie Åsberg, Jussi Jokinen

**Affiliations:** 1Department of Clinical Sciences/Psychiatry, Umeå University, 901 87 Umeå, Sweden; 2Department of Clinical Neuroscience, Karolinska Institute, 171 77 Stockholm, Sweden; 3Department of Clinical Sciences, Karolinska Institutet at Danderyd Hospital, 171 77 Stockholm, Sweden

**Keywords:** DNA Methylation, brain-derived neurotrophic factor, BDNF, suicide, emotionally unstable personality disorder, EUPD, borderline personality disorder, BPD

## Abstract

Brain-derived neurotrophic factor (*BDNF*) has previously been associated with the pathogenesis of both emotionally unstable personality disorder (EUPD) and suicidal behavior. No study has yet investigated *BDNF*-associated epigenetic alterations in a group of severely impaired EUPD and suicidal patients. The discovery cohort consisted of 97 women with emotionally unstable personality disorder (EUPD) with at least two serious suicide attempts (SAs) and 32 healthy female controls. The genome-wide methylation pattern was measured by the Illumina EPIC BeadChip and analyzed by robust linear regression models to investigate mean *BDNF* methylation levels in a targeted analysis conditioned upon severity of suicide attempt. The validation cohort encompassed 60 female suicide attempters, stratified into low- (n = 45) and high-risk groups (n = 15) based on degree of intent-to-die and lethality of SA method, and occurrence of death-by-suicide at follow-up. Mean *BDNF* methylation levels exhibited increased methylation in relation to EUPD (*p* = 0.0159, percentage mean group difference ~3.8%). Similarly, this locus was confirmed as higher-methylated in an independent cohort of females with severe suicidal behavior (*p* = 0.0300). Results were independent of age and BMI. This is the first study to reveal emerging evidence of epigenetic dysregulation of *BDNF* with dependence on features known to confer increased risk of suicide deaths (lethality of suicide-attempt method and presence of EUPD diagnosis with history of recent SAs). Further studies investigating epigenetic and genetic effects of *BDNF* on severe suicidal behavior and EUPD are needed to further elucidate the role of epigenetic regulatory mechanisms and neurotrophic factors in relation to suicide and EUPD, and hold potential to result in novel treatment methods.

## 1. Introduction

Emotionally unstable personality disorder (EUPD) is a psychiatric disorder characterized by an enduring pattern of affective instability, unstable relationships, efforts to avoid real or perceived abandonment, identity disturbance, and impulse-driven self-destructive behavior [1]. EUPD is a severely debilitating condition entailing a significant burden on those affected and their loved ones [2] and co-occur with other psychiatric disorders at a higher rate than any other personality disorder [3]. EUPD confers a high mortality rate with reported several-fold increases in both suicides and all-cause mortality in comparison to the general population [4]. EUPD has also been demonstrated to be associated with a wide array of non-psychiatric medical conditions that by themselves confer a high rate of morbidity and mortality [5]. Although EUPD patients consume healthcare at a high rate, they still have higher rates of health problems compared to the general population, possibly due to low compliance [6]. In addition to the burden of physical co-morbidity, EUPD is associated with a behavioral phenotype of poor health habits, i.e., smoking, physical inactivity, and rates of substance abuse of up to 50% [7]. EUPD is known to often co-occur with PTSD and around 30% of EUPD patients have PTSD [8].

EUPD is today thought to develop in an interplay between genetic predisposition and exposure to neglect and abuse in early life [9]. Several neurobiological mechanisms have been suggested as possible pathophysiological drivers of EUPD [10] and other personality disorders. A study measuring inflammatory factors in the plasma of patients with antisocial personality disorder with or without substance abuse disorder found an increase in pro-inflammatory factors and a heavy decrease in neurotrophic factors (including brain-derived-neurotrophic factor) regardless of the history of substance abuse, suggesting a possible systemic inflammatory mechanism and a lack of neuroprotective proteins underpinning the disorder [11]. The proposed relevance of dysregulation of the hypothalamic–pituitary–adrenal axis (HPA-axis) in EUPD has been extensively studied and emerging evidence implicates both epigenetic [12] and physiological [13] alterations in the condition. In a study of 49 patients with EUPD, the plasma levels of a wide array of enzymes involved in oxidative pathways were measured as well as several enzymes involved in anti-oxidative pathways. The study found a significant increase in inflammatory factors and a significant decrease in anti-inflammatory factors when compared to 33 healthy control subjects [14].

Early-life adversity (ELA)—an established risk factor of EUPD [9]—has been previously associated with increased circulating plasma levels of C-reactive protein in adults, a central component to systemic inflammation [15]. Furthermore, inflammatory marker interleukin 6 (IL-6), another protein central in inflammation, has been associated with high neuroticism, impulsivity [16], and low conscientiousness [17]—core traits of EUPD [18]. Attempts have been made to unify research pointing to childhood adversity with neurobiological models involving increased systemic inflammation and alteration in the HPA-axis [19,20]. Epigenetic changes of the genome with a resulting alteration in expression of both neuroprotective and inflammatory proteins have been suggested to play a role in EUPD, with previous research identifying methylation of *BDNF*-coupled promotor sites as particularly significant [21].

BDNF is a protein central in neurogenesis and the growth of neurons in the central nervous system. It supports the survival of neurons and is central also in their growth, differentiation, and plasticity throughout life. A decrease in *BDNF* expression has been linked to several psychiatric disorders including schizophrenia and depression. An absence of BDNF, as shown in knockout mouse models, leads to a failure of development and death shortly after birth. Decreased BDNF has been shown in layers IV and V of the dorsolateral prefrontal cortex—an area involved in working memory—in patients with schizophrenia spectrum disorder (SSD). Albeit inconclusive, previous studies have implicated that such effects may contribute to explaining working memory impairments seen in SSD. Furthermore, an inverse relationship between cortisol levels and BDNF has been found both in animal and human post-mortem studies of SSD [22]. Emerging evidence further implicates *BDNF* in the pathophysiology of depression. For example, rodents subjected to persistent stress were shown to exhibit decreased *BDNF* expression levels and hippocampal atrophy—an alteration that has also been evinced in major depressive disorder in humans. The relevance of *BDNF* in depression is further supported by studies demonstrating that antidepressant drugs are associated with a reversal of hippocampal atrophy and contemporary increases in *BDNF* expression levels [23]. A rat study exposed postnatal animals to stressed caretakers that displayed abusive behaviors. The maltreated rats displayed an increase in methylation in the *BDNF* gene encoding the protein. On the other hand, no differences in circulating *BDNF* were evinced in a systematic review of studies comparing whole-blood-derived gene expression levels of psychiatric patients with a history of suicidal attempts with non-suicidal controls [24]. The studies included did, however, not discriminate between severe and less severe suicide attempts, and the authors emphasized the need for rigorous case–control studies of suicidality and *BDNF* [25].

### Aims

The main goal of the present study was to investigate whether epigenetic dysregulation in *BDNF* was associated with EUPD and the severity of suicidal behavior (putatively with higher risk of later-completed suicide [26]) in females. To increase probabilities of deriving biologically meaningful results, we only considered methylation sites classified as differentially methylated regions (DMRs) and that were annotated to be located within 2000 bp of the transcriptional start site (TSS2000; a region where DNA methylation has been previously shown to exert larger influences on gene expression compared to other regions [27]). Using these stricter methylation site inclusion criteria, 16 *BDNF*-coupled sites were identified. To further increase robustness, the mean methylation value was calculated across methylation M-values of these 16 *BDNF*-coupled sites. As a first step, associations between candidate methylation sites and EUPD were investigated in a group of 97 females with emotionally unstable personality disorder (EUPD, previously borderline personality disorder) with severe suicidal behavior (a history of at least two recent suicide attempts) and 32 healthy volunteers. As a second step, candidate methylation markers were investigated in an independent group of 60 female suicide attempters, stratified into in silico generated high- and low-risk subgroups according to (a) a violent method of suicide attempt (i.e., hanging, shooting, or drowning) [28], (b) exhibiting a high intent-to-die as measured by the Freeman scale [29], or (c) having later died by suicide. These analyses pertained exclusively to female subjects.

## 2. Methods

### 2.1. Characterization of Discovery Group

The studied patient group consisted of 106 Caucasian females living in the Stockholm County, originally recruited as part of a randomized controlled trial (RCT) comparing two psychotherapy modalities (dialectical behavioral therapy and psychodynamic therapy) to treatment as usual (TAU) in women with EUPD with a history of severe suicidal behavior. Patients were referred to the study from all Stockholm County Council psychiatric clinics (a catchment area of 1.8 million inhabitants). Study inclusion criteria included a prior history of two or more potentially lethal suicide attempts (as defined by the patient’s belief that the attempt could have been lethal), with at least one attempt during the six months preceding referral. Subjects were excluded if presenting with a current life-threatening eating disorder, current psychotic disorder, or major depressive illness with melancholic features, evidence of dementia, or other irreversible organic brain syndrome or an active diagnosis of substance dependence [30]. The Structured Clinical Interview for DSM-IV Axis I and II interview (SCID-I and II) [31] schedules and the Comprehensive Psychopathological Rating Scale interview [32] were administered at baseline and EUPD (at the time, referred to as BPD) diagnosis, and psychiatric comorbidities were established after consensus diagnostic conference amongst experienced clinicians. Subjects were required to be in the 18–50-year-old age group. A total of 162 women with BPD were invited to take part in the SKIP project between 1999 and 2004. Of these individuals, 14 declined to join the study, 41 were excluded due to not fulfilling inclusion criteria or to fulfilling exclusion criteria, and one completed suicide before joining the study. Thus, out of 162 women, 106 (65%) Caucasian females took part in the SKIP study. The baseline measurements of this trial included—amongst others—a blood sample, which was analyzed to extract epigenetic data for the current project. Through the unique personal identification number in Sweden, patients were linked to the Cause of Death Register maintained by the Swedish National Board of Health and Welfare. Eight suicides were ascertained from the death certificates. Follow-up times ended in 2011.

#### 2.1.1. Ethics and Patient Consent

Details on the EUPD cohort have been previously published [30,33,34]. The original and the complementary study protocols were approved by the Committee for Ethical Research at Karolinska Institutet (Dnrs: 95–83; 2021-06929-01) and participants provided written informed consent.

#### 2.1.2. Blood Sample Collection, Methylation Profiling, and Data Processing (EUPD Group)

Standard procedures were utilized in blood sampling. Extraction from non-fasting subjects occurred in the morning. Retrieval of DNA from 97 EUPD participants and 32 control subjects were conducted by the phenol-chloroform method and samples were subjected to bisulfite conversion according to the EZ DNA Methylation GoldTM kit (ZymoResearch, Irvine, CA, USA). Hybridization of the resulting DNA to the Illumina Infinium Methylation EPIC BeadChip was executed. Thereafter, methylation values for 850 K probes for each sample were obtained through array imaging by an Illumina iScan system (Illumina, San Diego, CA, USA). Quality control and normalization of raw methylation IDAT data were conducted with the mefill package for R statistics (https://github.com/perishky/meffil/ accessed on 1 June 2022), utilizing control probes to isolate biological variations from technical variations. All samples passed QC, resulting in extraction of methylation β-values in a total of 129 subjects (97 EUPD and 32 control subjects).

#### 2.1.3. Annotation and Selection of DNA Methylation Probes

First, we identified 17 individual CpG-sites annotated to the gene *BDNF* that were classified as differentially methylated regions (DMRs) in the Illumina EPIC ManiFest annotation file. The DMR-regions were annotated to the following island shores: ‘Island’ (n = 11), N_Shore (n = 1), and S_Shore (n = 5). To increase the chance of biologically relevant results, we only considered CpG-sites annotated to be located within 2000 bp of the transcriptional start site—a region whereby the relationship between DNA methylation and gene expression have been previously shown to be more closely related compared to other regions [27]. Methylation loci situated more than 2000 base pairs away from the TSS were, thus, excluded from the subsequent analysis. One out of 17 *BDNF*-associated loci was situated more than 2000 base pairs from the TSS and was excluded, resulting in 16 individual CpG-sites included in the subsequent analysis. To increase the robustness of the derived values, individual CpG-sites were first transformed from betas into M-values, which have been shown to be statistically more robust [35]. Transformed values were subsequently aggregated by both mean and median values across these 16 *BDNF*-associated sites. The individual CpG-site values were plotted in boxplots together with derived mean and median values. From visual inspection of these boxplots for the EUPD and Suicide study groups, it was determined that the mean and median value were adequate representations of these individual values (Appendix A). The mean value was implemented for subsequent analyses in both datasets. Following these procedures, each subject was represented by one mean value derived across these 16 *BDNF*-coupled CpG-sites.

#### 2.1.4. DNA Methylation Association Study

Associations across M-transformed mean *BDNF*-methylation M-values and clinical groups (EUPD vs. Control) were investigated by robust linear regression models using the R-package ‘robustbase’ [36], specifying the recommended setting (KS2014). Chain-of-regression estimates included the standard MM-regression estimator (guaranteeing an acceptable compromise between high breakdown (i.e., 50%) and very high efficiency (i.e., 95%) [37]). As methylation levels have been previously shown to be heavily influenced by age and BMI, the analysis was adjusted for age and BMI. Models exhibiting *p*-values for the primary explanatory variable <0.05 were considered significant. As a post hoc analysis, individual M-transformed values for CpG-sites were individually investigated by robust linear regression models, stating the same specifications as in the main analysis. Thereafter, we analyzed whether the investigated CpG-sites exhibited an abundance of differentially methylated CpG-sites using binomial tests (as previously performed by Boström et al. [38]). *p*-value thresholds were set to the highest observed below a *p*-value of 0.05 to stratify probes according to significant and non-significant methylation changes. Subsequently, binomial tests were performed in R using the function ‘binom.test’, contrasting the number of nominally significant CpG-sites to the total number (n = 16) of investigated probes annotated as *BDNF* exhibiting methylation alterations in the same direction. Binomial test *p*-values were not adjusted for multiple testing, as they pertained to one gene only. A non-adjusted binomial test *p*-value < 0.05 was considered significant and indicative of an abundance of *BDNF*-associated probes differentially methylated in EUPD or the severe suicide attempt group.

### 2.2. Characterization of the Validation Dataset

#### 2.2.1. Ethics and Patient Consent

Details on the cohort of death by suicides and stratification of subjects have been previously published [26]. In brief, the study protocols were approved by the Regional Ethical Board in Stockholm, Sweden (Dnrs: 00-194, 2015/1454.32) and the participants gave their informed consent to the study, which was conducted in accordance with guidelines and regulations.

#### 2.2.2. Validation Cohort of Suicide Attempters

The study included adult patients that were clinically assessed in 2000–2005 at a Suicide Prevention Clinic (Karolinska university hospital) after conducting documented acts of self-destruction with variable intentions to die. Subjects with psychotic disorders, dementia, mental disability, or (intravenous) substance abuse were excluded. To improve the clinical generalizability of the study, non-suicidal self-injury and/or non-intravenous substance abuse were not included in the participant exclusion criteria. After exclusion of subjects declining to participate (n = 50), not meeting inclusion criteria (n = 61), or unable to attend clinical follow-up visits (n = 47), 100 patients were included in the study (33 men and 67 women). A total of 26 subjects were medication-free at the time of assessment. The medications reported in more than five individuals were antidepressants (Sertraline (n = 20), Citalopram (n = 12), Mirtazapine (n = 12), Venlafaxine (n = 9), and Fluoxetine (n = 7)). No patients received lithium treatment [26]. DNA samples were extracted from 88 of these subjects. The information about suicide and the cause thereof was available from matching unique identification numbers with the national Cause of Death register. Later death by suicide was established in four cases in the years 2004, 2006, 2007, and 2014, three of which were from hanging and one of which was from substance intoxication. Subjects that used a violent method of attempt but not death by suicide (i.e., hanging, shooting, or drowning) [28], exhibiting a Freeman scale of >6 [29], or having later died by suicide were classified as high-risk.

#### 2.2.3. Descriptive Statistics (Validation Group)

The classification of violent and non-violent suicide attempts into low- or high-risk groups has been previously described in detail [26]. In brief, dichotomization was performed with relevance concerning putative biological differences, and subjects fulfilling any of the following criteria were classified as high-risk: violent suicide attempt method, or a high score in the Freeman Scale, or later death by suicide. In accordance with the literature, non-violent attempts included substance intoxication or a single wrist-cut, whereas violent attempts pertained to all other available suicide methods, i.e., attempted drowning, shooting, gassing, several deep cuts, or hanging [28].

The Freeman scale captures key elements of suicide intent in assessing two primary objectives in relation to suicide attempts: interruption and reversibility probability. The inter-rater reliability of interruption and reversibility probability amounted to 0.8 and 0.97, respectively [29]. The second part of the scale measures likelihoods of interruption of attempted suicide by others whereby a high score exerts this as unlikely, conferring a higher risk of suicide completion. The reversibility scale assesses the probability of reversing the suicide attempt by, for example, assessing the amount and class of substance consumed or the extent of inflicted self-injury. Similarly, a lower reversibility probability is associated with a higher risk of death and confers a higher score on the Freeman scale. The subscales are rated 1–5 and combined for a total score of 2–10 [29]. Severe suicide attempts conferring a higher risk of death by suicide are inferred from scores >6. The discriminating validity of the scale was considered very good in a previous study consisting of a large sample of attempted-suicide subjects and suicide victims [29].

The Shapiro–Wilk’s test was implemented to assess skewness and kurtosis of the distribution of included clinical outcome variables, inferring a normal distribution (ND) for body-mass index (BMI) and age of participants. Independent samples *t*-tests were used to assess group differences for ND continuous variables, Mann–Whitney U-tests for non-ND continuous variables, and Fisher’s Exact test in the case of categorical variables.

#### 2.2.4. Blood Sample Collection, Methylation Profiling, and Data Processing (Validation Group)

Peripheral blood samples were collected according to standardized procedures. Study participants were required to be fasting at the time of blood sampling, which occurred in the morning. DNA was retrieved using the phenol-chloroform method, after which bisulfite conversion was performed using the EZ DNA Methylation GoldTM kit (ZymoResearch, Irvine, CA, USA). DNA specimens were thereafter hybridized to the Illumina Infinium Methylation EPIC beadchip and the array was imaged using the Illumina Iscan system (Illumina, San Diego, CA, USA), resulting in the quantification of methylation values at approximately 850,000 unique methylation identifiers across all samples. Preprocessing of methylation data included background correction, adjustment for methylation site measurement techniques (i.e., type I and type II probes), adjusting for any potential batch effects, and the removal of putatively unreliable probes. DNA methylation data were corrected for in silico generated surrogate measures of white blood cell type heterogeneity. Evaluation of sample outliers based on methylation data was performed by principal component analysis (PCA). Methylation preprocessing steps were performed using R software, version 3.3.0, and the following bioconductor packages—minfi [39], watermelon [40], sva [41], champ [42], and FactomineR [43]. A detailed description of the preprocessing steps has been previously published [26].

#### 2.2.5. DNA Methylation Association Study (Validation)

To investigate associations between mean *BDNF*-associated promoter methylation levels and severe clinical psychiatric phenotypes, we studied mean values across 16 *BDNF*-associated CpG-sites. In this analysis, as clinical variables were largely comparable between the in silico stratified groups, mean *BDNF* DNA-methylation M-values were compared between groups in using the one-sided Wilcoxon signed rank sum exact test, specifying the alternative hypothesis that methylation values are higher in severe suicide attempters (the same loci were greater-methylated in female EUPD patients). Post hoc univariate analyses were performed to exclude confounders from age and BMI on DNA methylation levels, correlating mean *BDNF* methylation levels with these descriptive variables in using Pearson’s moment correlation coefficients. *p*-values < 0.05 were considered significant.

## 3. Results

### 3.1. Promotor-Associated BDNF Is Higher-Methylated in Whole Blood of EUPD Patients (n = 129)

The 97 EUPD participants were all females with a mean age and BMI of 29.4 years (SD = 7.6) and 24.5 kg/m^2^ (SD = 4.7), respectively. Approximately a third had attained a university-level education and 10.3% had biological children. Active or previous tobacco usage was recorded in 57.7% of the EUPD-group. All subjects fulfilled criteria for one or more Axis I psychiatric diagnoses. Anxiety disorders were most commonly prevalent (60.8%) and a substantial proportion had major depressive disorder (MDD, 42.3%), of which a subset presented with severe MDD (13.4%). Naturally, all participants met criteria for EUPD, 32% of which fulfilled criteria for 7 or more items. Subjects had a mean global assessment of functioning (GAF) score of 49.2, corresponding to serious symptoms or serious impairment in social, occupational, or school functioning. History of alcohol and substance abuse was highly prevalent (33.0% and 26.8%, respectively). Benzodiazepines constituted the most frequent psychotropic medication (36.1%), closely followed by selective-serotonin reuptake inhibitors (SSRI; 33.0%), non-SSRI antidepressants (20.6%), neuroleptics (12.4%), and mood stabilizers (4.1%). All EUPD subjects had a history of recent suicide attempts (constituting inclusion criteria whereby the mean age at first suicide attempt was 20 years (SD = 7.5). Eight women (8.25%) died by suicide after the study was concluded, in the years 2002-2012. These deaths occurred by intoxication (N = 5), hanging (N = 2), and railway suicide (N = 1). One additional subject died of unknown cause, which was not categorized as a suicide-related death (Table 1).

In the association analysis between whole-blood-derived BDNF-DNA methylation and EUPD, we studied mean values across 16 *BDNF*-associated promoter CpG-sites. Mean and median-derived values did not significantly differ, as measured by unpaired independent samples *t*-tests (*p* = 0.661), and the mean value was determined upon visual inspection as representative of the underlying data (Appendix A). In the subsequent analysis—implementing robust linear regression models adjusted for age and BMI—mean BDNF DNA-methylation M-values were increased with dependence on EUPD (compared to controls) (*p* = 0.0159). Across all samples, non-M-transformed mean BDNF methylation beta-values were 0.056 (SD = 0.0059) with a mean between-group difference of ~3.8% (Table 2 and Figure 1).

As a post hoc sensitivity analysis, we re-performed the above analysis for the 16 individual CpG-sites separately and evaluated whether there was an abundance of individual *BDNF*-associated probes differentially methylated in EUPD. Three probes exhibited higher methylation with dependence on EUPD (i.e., cg11718030, *p*-value = 0.0002; cg15462887, *p*-value 0.0434; cg24377657, *p*-value = 0.0057). The number of nominally significant probes (n = 3) were contrasted to the total number of probes (n = 16) by the highest nominally significant p-value observed (i.e., *p* = 0.0434) in an exact binomial test. This analysis yielded a *p*-value of 0.0299, indicative of an abundance of individual *BDNF*-associated probes differentially methylated in EUPD (Table 3).

### 3.2. Cohort Description (Validation Group)

All participants were females (males were excluded). A total of 15 patients (25%) were classified into the high-risk/severe attempt category and 45 (75%) into the low-risk group. Clinical outcome variables were largely comparable between the two groups (*p* > 0.05), i.e., BMI, psychiatric diagnoses, exposure to violence in childhood or adulthood [26], alcohol dependence, or substance abuse (Table 4).

### 3.3. Promotor-Associated BDNF-Methylation Levels Are Higher-Methylated with Dependence on Severity of Suicide Attempt (n = 60)

In the association analysis between whole-blood-derived *BDNF*-DNA methylation and severity of suicide attempt, we studied mean values across 16 *BDNF* promoter-associated CpG-sites. Mean and median-derived values did not significantly differ, as measured by unpaired independent samples *t*-tests (*p* = 0.939), and the mean value was determined upon visual inspection as representative of the underlying data (Appendix A). DNA methylation levels were higher-methylated with dependence on the severity of suicidal behavior (compared to lower-risk suicide attempters) (W = 623, *p* = 0.0231) (n = 60) (Figure 2).

Post hoc analyses by Pearson’s product moment correlation coefficient yielded that derived DNA methylation levels were unassociated with age and BMI (*p* > 0.1). Across all samples, non-M-transformed mean BDNF methylation beta-values were 0.0689 (SD = 0.0305) with a mean between-group difference of ~2.0% (Figure 2). As a post hoc sensitivity analysis, we re-performed the above analysis for the 16 individual CpG-sites separately and evaluated whether there was an abundance of individual *BDNF*-associated probes differentially methylated in severe suicidal behavior. Two probes exhibited higher methylation with dependence on EUPD (i.e., cg15462887, *p*-value = 0.005; cg23497217, *p*-value = 0.002). The number of nominally significant probes (n = 2) were contrasted to the total number of probes (n = 16) by the highest nominally significant p-value observed (i.e., *p* = 0.005) in an exact binomial test. This analysis yielded a p-value of 0.00286, indicative of an abundance of individual *BDNF*-associated probes differentially methylated in severe suicidal behavior (Table 5).

## 4. Discussion

This study reports on a targeted DNA methylation analysis of *BDNF*-coupled CpG-sites in a cohort of 97 severely impaired EUPD patients with a history of suicide attempts and 32 non-psychiatric controls. Validation of these findings was investigated in an independent cohort of 60 female suicide attempters, stratified into high and low risk based on the severity of suicide attempts. In these analyses, mean *BDNF* methylation levels were higher-methylated in relation to EUPD and severity of suicidal behavior—independent of age and BMI. This is the first study to reveal emerging evidence of *BDNF*-coupled increased methylation with dependence on EUPD in women with severe suicidal behavior. These results are strengthened by the validation—of similar magnitude and direction—in a wholly independent cohort of women with dependence on the severity of suicidal behavior. These findings could also be interpreted as more robust compared to studies investigating individual CpG-sites, by accounting for the mean value across several putatively inter-linked *BDNF*-coupled methylation loci. Thus, epigenetic alterations in *BDNF* are associated with both EUPD and severe suicidal behavior in female subjects. Causal inferences are prevented by the cross-sectional study design, which did not allow for the determination of whether the observed alterations contribute to the pathophysiology of the condition or arise as an effect of the condition. Nevertheless, extensive previous research on the topic of *BDNF* and psychiatric disorders adds to the relevance of these findings. According to classical models of epigenetic regulatory mechanisms, increased DNA methylation inhibits gene transcription, implicating, although not directly studied in this population, that decreased *BDNF*-levels may be associated with EUPD and severity of suicidal behavior. Targeted studies measuring both DNA methylation and gene expression levels in the brain of concerned subjects are needed to further elucidate the role of *BDNF* in EUPD and suicidality.

Strengths of the study include two phenotypically well-defined, representative patient cohorts of suicide attempters with thorough diagnostics of the psychiatric disorders and a careful assessment of the severity of suicidal behavior [26]. In addition, possible confounders such as psychotropic medication usage, childhood adversity, and comorbidity patterns were considered. It is clear that suicide attempts are highly heterogeneous and prospective studies demonstrate that high intent to die and choice of suicide method evince a higher risk of death by suicide [26,28,44]. Consequently, research suggests that suicide victims are more similar to high-intent or violent attempters as compared to non-violent attempters [28]. It can, thus, be argued that a lack of deep phenotyping and/or stratification of suicide attempters may prevent assured conclusions from previous studies exploring this field [45]. Moreover, the study pertained only to promotor-associated DNA methylation markers measured in whole blood deemed more influential on gene expression compared to other DNA-methylation regions, and the differential methylation status of the candidate site was consistently higher-methylated across the two investigated cohorts with dependence on phenotypes sharing similar features (i.e., severe suicidality).

Our study is burdened by several limitations. First, given the relatively small sample size, the main analysis would be underpowered to comprehensively detect subtle changes in DNA methylation. Yet, lower-powered studies are appropriate in detecting group-differences with a larger effect size, lending further support for the relevance of the presented findings. Second, as derived methylation changes were small (i.e., mean absolute difference in methylation beta-value of 0.06–3.6%), the biological relevance of such subtle changes could be questioned. Recent studies, however, implicate that methylation changes in the 1–5% range are associated with extensive transcriptional and translational consequences—a magnitude of alteration deemed particularly relevant in the pathophysiology of complex multifactorial psychiatric conditions [46]. Third, whole-blood-derived *BDNF* methylation levels were higher-methylated with dependence on EUPD and severity of suicide attempt. While this cohort of severely impaired EUPD patients with a history recent of suicide attempts could be argued to exhibit similar features to the cohort assessing the severity of suicidal behavior, replication of our findings in relation to EUPD or severe suicidal behavior would be of value to discern whether the observed alterations are related to EUPD or suicidality (or both). Fourth, the appropriateness of the assumption of blood–brain transferability of epigenetic alterations has come under increasing scrutiny [47]. Comparing individual Illumina-array-derived methylation samples extracted from whole blood and several brain regions in 122 individuals, Hannon et al. showed that, for a majority of epigenetic sites, blood-based alterations provide limited causative information as to disease-inducing epigenetic changes occurring in brain tissue [47]. Thus, the uncritical extrapolation of our findings to putative alterations in brain tissue is cautioned against. The aim of the present study was to explore any associations between epigenetic markers of BDNF in whole blood and severe clinical psychiatric phenotypes. Further studies are needed to elucidate any downstream effects of the observed results on gene expression in brain tissue. Fifth, this study pertained exclusively to female subjects. Thus, no assured conclusions can be drawn in relation to *BDNF* blood methylation levels in males with EUPD or severe suicidal behavior. Sixth, given the limited scope of the study objectives—to investigate blood-derived DNA methylation markers in relation to EUPD and severity of suicide attempt—genotyping of *BDNF* was not performed. Future studies investigating associations between *BDNF* polymorphisms and promotor methylation levels are needed to fully elucidate whether putative underlying pathophysiological mechanisms are conferred by epigenetic, genetic, or combined effects. Seventh, associations between methylation levels and gene expression levels were not investigated. This was a conscious choice of the authors as the putative prevalence of driving *BDNF* gene mutations would need to be accounted for the adequate interpretation of such analyses [48]. Moreover, openly accessible data were not available for the examination of DNA methylation and gene expression in representative cohorts of severely impaired EUPD patients or suicide attempters. Thus, the adjunctive value of investigating methylation-expression correlations in whole blood of control subjects was considered of marginal relevance and was, thus, not explored. Lastly, the DNA methylation analysis did not account for potential confounders from non-antidepressant medication use. Antidepressants were the only medication class reported in more than five subjects and there were no significant between-group differences in any disorders precipitating the prescription of such medications (i.e., MDD or anxiety disorders). In addition, the use of SSRI medication was independent of *BDNF* methylation levels in the EUPD group. Thus, while it cannot be completely excluded, substantial bias from medication use to the present findings would be highly unlikely. Lastly, we restrict pre-specified inclusion criteria to only include CpG-sites annotated as differentially methylated regions (DMRs) and located within 2000 base pairs of the transcriptional start site. These criteria precluded the investigation of other genes that have previously been associated with neuroinflammation. For example, CRP, IL-1, IL-2, and IL-6 were not annotated to any DMR probe within the TSS2000 and were, thus, not investigated.

In conclusion, whole-blood-derived *BDNF promotor* methylation levels are higher-methylated with dependence on EUPD and severity of suicide attempts in females. Ample research previously implicated *BDNF* gene variations in relation to suicide deaths and EUPD, arguably providing strong support to the relevance of the presented findings. Future studies performed in brain tissue, males, or investigating associations between *BDNF* polymorphisms and such methylation levels are needed to fully elucidate whether the observed associations contribute to major underlying pathophysiological mechanisms in EUPD and/or severe suicidal behavior. Results of such studies hold potential to aid efforts aimed at preventing suicide and managing EUPD by granting complementary insights into the underlying pathophysiological processes and could aid in advancing novel treatments.

## Figures and Tables

**Figure 1 cells-12-00350-f001:**
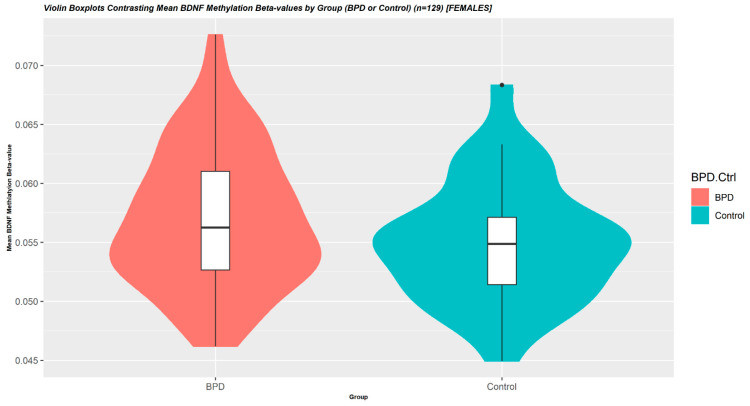
Violin Boxplots Contrasting Mean BDNF Methylation Beta-values by Group (EUPD or Control) (n = 129) [FEMALES]. On the association analysis between whole-blood-derived *BDNF*-DNA methylation and EUPD, we studied mean values across 16 *BDNF*-associated promoter CpG-sites. In this analysis—implementing robust linear regression models adjusted for age and BMI—mean *BDNF* DNA-methylation was increased with dependence on EUPD (compared to controls) (*p* = 0.0159). Across all samples, non-M-transformed mean *BDNF* methylation beta-values were 0.056 (SD = 0.00595) with a mean between-group difference of ~3.8%.

**Figure 2 cells-12-00350-f002:**
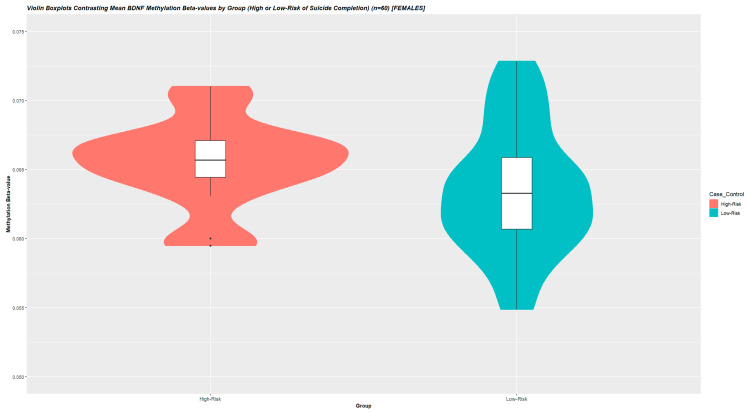
Violin Boxplots Contrasting Mean BDNF Methylation Beta-values by Group (High or Low-Risk of Suicide Completion) (n = 60) [FEMALES]. To investigate associations between mean *BDNF*-associated promoter methylation and severe clinical psychiatric phenotypes, we studied mean methylation M-values across 16 *BDNF*-associated promoter CpG-sites. Beta-values were used for illustration. In this analysis, as clinical variables were largely comparable between the in silico stratified groups, mean *BDNF* DNA-methylation was compared between groups using the one-sided Wilcoxon signed rank sum exact test. Post hoc univariate analyses were performed to exclude confounders from age and BMI on DNA methylation levels, correlating mean *BDNF* methylation levels with these descriptive variables in using Pearson’s moment correlation coefficients. *p*-values < 0.05 were considered significant. DNA methylation levels were higher-methylated with dependence on severity of suicidal behavior (compared to lower-risk suicide attempters) (W = 227, *p* = 0.0299) (n = 60). Across all samples, non-M-transformed mean BDNF methylation beta-values were 0.06885 (SD = 0.0306) with a mean between-group difference of ~2.0%.

**Table 1 cells-12-00350-t001:** Characteristics of subjects (EUPD grp).

	BPD	Control	Statistics (*t*-Test, Chisq-Test, Fisher’s Exact Test), *p*-Value
N	97	32	
Age (years), mean (SD)	29.4 (7.6)	37.2 (6.0)	**<0.0001**
Men:women, n (%)	0 (0.0):97 (100.0)	0 (0.0):32 (100.0)	*ns*
Biological children, n (%)	10 (10.3)	21 (65.6)	**<0.0001**
University Education, n (%)	28 (28.9)	N/A	-
BMI, mean (SD)	24.5 (4.7)	24.1 (3.7)	*ns*
Tobacco usage, n (%)	56 (57.7)	13 (40.6)	0.075
Active Major Depressive Disorder, n (%)	41 (42.3)	0 (0.0)	-
Severe MDD, n (%)	13 (13.4)	0 (0.0)	-
Bipolar Disorder II or UNS, n (%)	8 (8.2)	0 (0.0)	-
Comorbid Anxiety Disorder, n (%)	59 (60.8)	0 (0.0)	-
Borderline personality disorder, n (%)	97 (100.0)	0 (0.0)	-
- ≥7 fulfilled BPD criteria, n (%)	31 (32.0)	0 (0.0)	
History of alcohol abuse, n (%)	32 (33.0)	0 (0.0)	
History of substance abuse, n (%)	26 (26.8)	0 (0.0)	-
KIVS exposure to violent behavior during childhood (6–14 years of age), mean (SD)	2.66 (1.86)	0.36 (0.71)	**<0.0001**
KIVS exposure to violent behavior as adult (>15 years of age), mean (SD)	2.47 (1.90)	0.35 (0.76)	**<0.0001**
Global Assessment of Functioning (GAF), mean (SD)	49.2 (12.4)	N/A	-
Psychotropic Medication, n (%)			
SSRI	32 (33.0)	0 (0.0)	-
Non-SSRI antidepressants	20 (20.6)	0 (0.0)	-
Mood stabilizers	4 (4.1)	0 (0.0)	-
Benzodiazepines	35 (36.1)	0 (0.0)	-
Neuroleptics	12 (12.4)	0 (0.0)	-
History of past suicide attempt, n (%)	97 (100.0)	0 (0.0)	-
Age at first suicide attempt, mean (SD)	20.0 (7.5)	-	-
Later confirmed death by suicide, n (%)	8 (8.25)	0 (0.0)	0.20

*p*-values were calculated by means of *t*-test, Mann–Whitney U-test, or chi-square test, contrasting values for control and BPD subjects. Pre-specified group differences were not subjected to such analyses, i.e., frequency of psychiatric diagnoses, psychotropic medications, or history/age at first suicide attempt. Subjects with extensive past or ongoing tobacco usage were classified as users, whereas occasional/party smokers (and non-users) were categorized as non-users. History of alcohol or substance abuse was considered present when the diagnosis of an alcohol or substance abuse diagnosis was recorded (including cases where these were in remittance). Non-SSRI antidepressant in the majority pertained to ongoing usage of psychotropic drugs with active substances desvenlafaxine (Effexor), desmethylmirtazapine (Remeron/Mirtazapine), and Amitriptyline (Clomipramine)(in order of frequency). Mood stabilizers included Valproic Acid (Ergenyl), Litium (Lithionit), and Lamictal (Lamotrigine). Benzodiazepine medications used were alprazolam (Xanor), clomethiazole (Heminevrin), Oxazepam (Sobril), and Diazepam (Stesolid); Z-drugs Zopiclone (Imovane) and Zolpidem (Stilnoct) were also included. Neuroleptics included Levomepromazine (Nozinan), Risperidone (Risperdal), Olanzapine (Zyprexa), Haloperidol (Haldol), and Quetiapine (Seroquel). The number of recorded suicidal/parasuicidal events did not take severity of attempt into account (i.e., intoxications or self-cutting were weighted equally to more severe attempts involving, for example, violence). A one-tailed *p*-value < 0.05 was considered significant. Abbreviations: BMI, body-mass index; BPD, borderline personality disorder; KIVS, Karolinska Interpersonal Violence Scale; *ns*, not significant; SSRI, selective serotonin reuptake inhibitor.

**Table 2 cells-12-00350-t002:** Robust Linear Regression Contrasting Mean *BDNF* Methylation Levels to Disease Status (EUPD or Control).

Statistic	Coef.	Std. Error	*t* Value	*p*-Value
Intercept	−4.239911	0.1125374	−37.676	**<2 × 10^−16^**
EUPD vs. Control	0.0838778	0.0391741	2.141	**0.0343**
Age	0.0029001	0.0021716	1.335	*ns*
BMI	0.0002664	0.0035534	0.075	*ns*

Methylation beta-values across 16 CpG-sites (TSS2000, DMR) were averaged for each study participant. These mean *BDNF* methylation levels (mean values converted to M-values) were contrasted to disease status (EUPD (n = 97) vs. Control (n = 32)), adjusting for age and BMI using robust linear regression models with ‘MM’-estimators. Robust residual standard error: 0.1685. Adjusted R-squared: 0.01765. *p*-values < 0.05 were considered significant (**bold**). **Abbreviations**: BMI, body mass index; Coef., coefficient; EUPD, emotionally unstable personality disorder.

**Table 3 cells-12-00350-t003:** Robust Linear Regression Contrasting Individual BDNF Methylation M-values to Disease Status (EUPD or Control).

CpG	ProbeType	CHR	Pos	Location	Island_Shore	DMR	Coef.	*p*-Value
cg03167496	II	11	27743619	TSS200;TSS1500;TSS1500; TSS1500	Island	Yes	0.116	*ns*
**cg05218375**	**II**	**11**	**27723218**	**TSS1500;Body;5’UTR;TSS1500;5’UTR;5’UTR;TSS200;TSS1500;TSS1500;TSS1500;TSS1500;5’UTR;TSS1500;TSS1500**	**S_Shore**	**Yes**	**−0.132**	**0.0390**
cg06046431	I	11	27744490	TSS1500	Island	Yes	0.028	*ns*
cg06816235	II	11	27742219	Body;5’UTR;TSS1500;5’UTR;1stExon;1stExon;1stExon;5’UTR	Island	Yes	0.001	*ns*
cg06991510	I	11	27723237	TSS1500;Body;5’UTR;TSS1500;5’UTR;5’UTR;TSS200;TSS1500;TSS1500;TSS1500;TSS1500;5’UTR;TSS1500;TSS1500	S_Shore	Yes	−0.013	*ns*
cg10022526	II	11	27744557	TSS1500	Island	Yes	−0.022	*ns*
**cg11718030**	**II**	**11**	**27744363**	**TSS1500**	**Island**	**Yes**	**0.328**	**0.0002**
cg14589148	I	11	27743648	TSS200;TSS1500;TSS1500;TSS1500	Island	Yes	0.073	*ns*
**cg15462887**	**II**	**11**	**27744049**	**TSS1500**	**Island**	**Yes**	**0.144**	**0.0434**
cg16257091	II	11	27743580	TSS1500;TSS1500;1stExon;5’UTR;TSS1500	Island	Yes	−0.130	*0.0857*
cg22288103	II	11	27743654	TSS1500;TSS1500;TSS1500;TSS200	Island	Yes	0.109	0.0791
cg23497217	II	11	27723214	TSS1500;Body;5’UTR;TSS1500;5’UTR;5’UTR;TSS200;TSS1500;TSS1500;TSS1500;TSS1500;5’UTR;TSS1500;TSS1500	S_Shore	Yes	0.079	*ns*
**cg24377657**	**I**	**11**	**27723245**	**TSS1500;Body;5’UTR;TSS1500;5’UTR;5’UTR;TSS200;TSS1500;TSS1500;TSS1500;TSS1500;5’UTR;boldTSS1500;TSS1500**	**S_Shore**	**Yes**	**0.134**	**0.0057**
cg25156688	II	11	27744054	TSS1500	Island	Yes	0.108	*ns*
cg25381667	I	11	27743651	TSS200;TSS1500;TSS1500;TSS1500	Island	Yes	0.053	*ns*
cg26840770	II	11	27723290	TSS1500;Body;5’UTR;TSS1500;5’UTR;5’UTR;TSS200;TSS1500;TSS1500;TSS1500;TSS1500;5’UTR;TSS1500;TSS1500	S_Shore	Yes	0.120	*ns*

Results of post hoc analyses, investigating M-transformed CpG-sites individually by robust linear regression models, specifying the standard MM-regression estimator and adjusting for age and BMI. Probe annotations according to Price et al. are shown, including the Illumina type (I or II), the annotated chromosome, the exact probe position, and whether the probe is located on an island or shore and is a differentially methylated region. Coefficients and p-values represent the beta-value and significance level for each individual CpG-site, respectively. Nominally significant CpG-sites are highlighted in bold. Subsequently, the number of nominally significant probes (*p* < 0.05) with a positive beta-value (the same direction as the main analysis) (n = 3) were contrasted to the total number of investigated probes by one-sided binomial tests, specifying p-value thresholds at 0.05 to stratify probes according to successes (n = 3; total n = 16)—resulting in a *p*-value of 0.04924 (**not shown in table**), indicative of an abundance of *BDNF*-associated probes differentially methylated in EUPD. **Abbreviations**: CHR, chromosome; Coef., regression coefficient; DMR, differentially methylated region (yes or no), Island_Shore, annotated probe region (Island or Shore); Location, annotated probe region (Exon/Intron/TSS1500/2000); Pos, position; *p*-value, regression significance level; TSS1500, probe located within 1500 base pairs of the transcriptional start site; TSS2000, probe located within 2000 base pairs of the transcriptional start site.

**Table 4 cells-12-00350-t004:** Characteristics of subjects (Suicide grp).

	Attempted Suicide (n = 60)		
	High-Risk Group	Low-Risk Group	Statistics (*t*-Test, Mann–Whitney U-test, Chisq. Test, Fisher’s Exact Test), *p*-Value
N	15	45	
Age (years)	35.67 (13.2)	34.0 (12.4)	*ns*
BMI, mean (SD)	24.3 (4.6)	24.9 (4.3)	*ns*
Borderline personality disorder, n(%)	3 (20.0)	5 (11.1)	*ns*
Other personality disorder, n(%)	3 (20.0)	8 (17.8)	*ns*
Alcohol dependence, (n(%))	4 (26.7)	7 (15.6)	*ns*
Substance dependence, n(%)	1 (6.7)	6 (13.3)	*ns*
Completed suicide, n(%)	1 (6.7)	0 (0.0)	*ns*
KIVS subscale, n(%)			
Exposure violent behavior during			
Childhood	5 (33.3)	14 (31.1)	*ns*
Adulthood	6 (40.0)	13 (28.0)	*ns*

Values are shown as mean (SD) unless otherwise specified. *p*-values were calculated by means of *t*-test, Mann–Whitney U-test, chi-squared test, or Fisher’s exact test (when the expected numbers per cell were below 5), contrasting values for subjects in the high-risk vs. low-risk suicide attempt group. A one-tailed *p*-value <0.05 was considered significant. Abbreviations: KIVS, Karolinska Interpersonal Violence Scale; *ns*, not significant.

**Table 5 cells-12-00350-t005:** Wilcoxon Rank Sum Exact Test Contrasting Individual BDNF Methylation M-values to Severity of Suicidal Behavior.

CpG	ProbeType	CHR	Pos	Location	Island_Shore	DMR	Coef. (W)	*p*-Value
cg03167496	II	11	27743619	TSS200;TSS1500;TSS1500;TSS1500	Island	DMR	323	*ns*
cg05218375	II	11	27723218	TSS1500;Body;5’UTR;TSS1500;5’UTR;5’UTR;TSS200;TSS1500;TSS1500;TSS1500;TSS1500;5’UTR;TSS1500;TSS1500	S_Shore	DMR	255	0.082
cg06046431	I	11	27744490	TSS1500	Island	DMR	346	*ns*
cg06816235	II	11	27742219	Body;5’UTR;TSS1500;5’UTR;1stExon;1stExon;1stExon;5’UTR	Island	DMR	266	ns
cg06991510	I	11	27723237	TSS1500;Body;5’UTR;TSS1500;5’UTR;5’UTR;TSS200;TSS1500;TSS1500;TSS1500;TSS1500;5’UTR;TSS1500;TSS1500	S_Shore	DMR	364	*ns*
cg10022526	II	11	27744557	TSS1500	Island	DMR	362	ns
cg11718030	II	11	27744363	TSS1500	Island	DMR	321	*ns*
cg14589148	I	11	27743648	TSS200;TSS1500;TSS1500;TSS1500	Island	DMR	350	ns
**cg15462887**	**II**	**11**	**27744049**	**TSS1500**	**Island**	**DMR**	**187**	**0.005**
cg16257091	II	11	27743580	TSS1500;TSS1500;1stExon;5’UTR;TSS1500	Island	DMR	322	ns
cg22288103	II	11	27743654	TSS1500;TSS1500;TSS1500;TSS200	Island	DMR	328	*ns*
**cg23497217**	**II**	**11**	**27723214**	**TSS1500;Body;5’UTR;TSS1500;5’UTR;5’UTR;TSS200;TSS1500;TSS1500;TSS1500;TSS1500;5’UTR;TSS1500;TSS1500**	**S_Shore**	**DMR**	**169**	**0.002**
cg24377657	I	11	27723245	TSS1500;Body;5’UTR;TSS1500;5’UTR;5’UTR;TSS200;TSS1500;TSS1500;TSS1500;TSS1500;5’UTR;TSS1500;TSS1500	S_Shore	DMR	329	*ns*
cg25156688	II	11	27744054	TSS1500	Island	DMR	246	0.061
cg25381667	I	11	27743651	TSS200;TSS1500;TSS1500;TSS1500	Island	DMR	382	ns
cg26840770	II	11	27723290	TSS1500;Body;5’UTR;TSS1500;5’UTR;5’UTR;TSS200;TSS1500;TSS1500;TSS1500;TSS1500;5’UTR;TSS1500;TSS1500	S_Shore	DMR	395	*ns*

Results of post hoc analyses, investigating M-transformed CpG-sites individually by the univariate Wilcoxon Rank Sum Exact Test, specifying the alternative hypothesis that methylation values are higher in severe suicide attempters. Probe annotations according to Price et al. are shown, including the Illumina type (I or II), the annotated chromosome, the exact probe position, and whether the probe is located on an island or shore and is a differentially methylated region. Coefficients and p-values represent the test statistic (Wilcoxon ‘W’) and significance level for each individual CpG-site, respectively. Nominally significant CpG-sites are highlighted in bold. Subsequently, the number of nominally significant probes (*p* < 0.05) exhibiting greater methylation in severe suicide attempters (i.e., the same direction as the main analysis) (n = 2) were contrasted to the total number of investigated probes by one-sided binomial tests, specifying p-value thresholds at the highest observed below a *p*-value of 0.05 (i.e., cg15462998-0.005) to stratify probes according to successes (n = 2; total n = 16)—resulting in a *p*-value of 0.00286 (**not shown in table**), indicative of an abundance of *BDNF*-associated probes differentially methylated in severe suicide attempters. **Abbreviations**: CHR, chromosome; Coef., regression coefficient; DMR, differentially methylated region (yes or no), Island_Shore, annotated probe region (Island or Shore); Location, annotated probe region (Exon/Intron/TSS1500/2000); Pos, position; P-value, regression significance level; TSS1500, probe located within 1500 base pairs of the transcriptional start site; TSS2000, probe located within 2000 base pairs of the transcriptional start site.

## Data Availability

The data underlying the findings presented in this study are available upon reasonable request.

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
