# Peer review of "Increased Methylation of Brain-Derived Neurotrophic Factor (BDNF) Is Related to Emotionally Unstable Personality Disorder and Severity of Suicide Attempt in Women"

_cells, 2023, doi:10.3390/cells12030350_

Round 1

Reviewer 1 Report

The manuscript  “Epigenetic Hypermethylation of Brain-Derived Neurotrophic Factor (BDNF) is related to Emotionally Unstable Personality Disorder and Severity of Suicide Attempt in Women” describes  a study testing the association of  methylation levels of BDNF, CRP, IL-1, IL-2 and IL-6 with EUPD and suicidal behavior and its severity in a discovery and a validation set of samples . The discovery group used a case-control design of EUPD patient vs.  healthy controls,  while for the all suicide attempter validation cohort, the research team used their previously published criteria to classify suicide attempters into high and low risk.

The study contains some methodological steps that are novel and are designed to increase power by pre- aggregating estimates. However, it is not clear that these steps are based on valid reasoning (see below), and the authors should either  present more evidence as to the validity or modify the aggregation method.

Otherwise, this is a well-written and interesting study with scientific merit, and the use of separate discovery and validation samples is a considerable advantage, The team has experience in this research area, and with some more work this paper could be a nice addition to the scientific literature on the biological basis of personality disorders and suicidal behavior.

Specific comments:

1. The authors’ stated reason for aggregating data across several loci is one of robustness. They find 16 loci on the BDNF differentially methylated  (associated with the outcome), then take the average of the beta values , and claim that  this is more robust, then using   then transform that to an M value as it is “more robust”. However, there are two problems here. First, means (as opposed to medians) are well-known  to not be robust to outliers - this is not a matter of outliers in methylation values, which were dealt with in  are-processing step, rather it is about  outliers among the loci-specific beta estimates- so evidence should be presented of their distribution before claiming robustness of the mean.

The second issue is that the transformation from the beta to the M is not linear, which means that the average of  betas would not get transformed into the mean M value. This means that the transformed M value has no clear meaning. It seems that a better process would be to first transform the betas into M values and then aggregate. Why was this not the method chosen?

2.  The loci were first tested for  differential methylation, and then one of the DMRs was excluded for being more than 2000 base pairs away from the transcriptional start site. If such “far” loci were not of scientific interest, why were they included in the analysis at all? It would make more sense to screen them out at pre-processing time, but even if they are included in the analysis for ease, they should be screened out before the p-values are evaluated for significance.

3. Chi-square tests seem to have been used in Table 3 for binary variables even when the expected numbers per cell were below 5- Fisher’s exact test should be used instead.

Reviewer 2 Report

The manuscript by Jamshidi et al. assesses, by means of Illumina EPIC BeadChip, methylation of 16 CpGs in BDNF gene promoter in whole blood samples obtained from  97 women with emotionally unstable personality disorder (EUPD) and 32 healthy female controls. The Authors find a 3.8% higher mean methylation value in EUPD subjects. They further validate the results in a cohort of low- and high-risk suicide attempters and find that the methylation is higher in women with higher suicidal risk. They identify a correlation  between BDNF promoter methylation and the suicidal risk and conclude that BDNF promoter methylation may contribute to the pathophysiology of EUPD and suicidal behavior.

I have several critical comments on the manuscript.

1. Why did the Authors present only the mean methylation value of the 16 differentially methylated CpGs? Results for individual cytosines should could also be interesting. In any case, the genomic coordinates of the investigated cytosine residues should be given.

2. The description in “Aims” and Paragraph 2.1.5., concerning genes other than BDNF , is not  clear and the sentence in Aims “Thus, CRP, IL-1, IL-2 and IL-6 could not be reliably investigated” is enigmatic. If the methylation analysis did not identify any differences in CpG methylation within  2000 bp of TSS of those genes, this should be clearly stated; if, on the other hand, results on their methylation could not be obtained due to technical or other problems those genes should not be included in the manuscript.

3 . I am not sure if the prefix “hyper” should be used in the  context of methylation results described in this manuscript since 1) the overall methylation level of the 16 differentially methylated CpGs in BDNF gene promoter, judging from the beta values in Fig. 1, is very low (below 10%) and 2) the difference in the mean methylation value between control and EUPD subjects is only 3.8% .  “Increased/higher methylation” seems to better describe the results.

2. There are many awkward phrases stating that “level”, “status” or even “methylation” are/were “hypermethylated” eg., “ differential methylation status of the candidate site was consistently hypermethylated”; “BDNF-methylation Levels are Hypermethylated”; “According to classical models of epigenetic regulatory mechanisms, DNA methylation hypermethylation inhibits gene translation”.  Also, regarding the last sequence, DNA methylation inhibits transcription, not translation.

3. The sentence ““This study reports on a targeted DNA methylation analysis of BDNF-coupled CpG-sites in a cohort of 129 severely impaired EUPD patients with a history of suicide attempt” gives a different number of EUPD patients (129) than it is stated in the Abstract (97) and elsewhere in the text.

Minor:

1. The description of y-axis in Fig.2 is illegible.

2. The use of  italic for BDNF gene (and other genes) is not always consistent, e,g. page 8 “BDNF promoter”.

3. “Promotor Associated BDNF is higher methylated” should rather be “ BDNF (associated) promotor is more highly methylated” .

4. Paragraphs 2.1.3 and 2.1.5 are largely repetitive.

5. DMS (s=site) is a more appropriate term than DMR to use in this study.

Round 2

Reviewer 1 Report

The authors have been very responsive to all the points I made, and the manuscript is improved as a result.  I have no further comments to make